# Adaptive Multiclass Mahalanobis Taguchi System for Bearing Fault Diagnosis under Variable Conditions

**DOI:** 10.3390/s19010026

**Published:** 2018-12-21

**Authors:** Ning Wang, Zhipeng Wang, Limin Jia, Yong Qin, Xinan Chen, Yakun Zuo

**Affiliations:** 1State Key Lab of Rail Traffic Control and Safety, Beijing Jiaotong University, Beijing 100044, China; 17114235@bjtu.edu.cn (N.W.); yqin@bjtu.edu.cn (Y.Q.); 15114217@bjtu.edu.cn (X.C.); 18120772@bjtu.edu.cn (Y.Z.); 2National Engineering Laboratory for System Safety and Operation Assurance of Urban Rail Transit, Guangzhou 510000, China; 3Beijing Research Center of Urban Traffic Information Sensing and Service Technologies, Beijing Jiaotong University, Beijing 100044, China

**Keywords:** fault diagnosis, bearing, SVD, VMD, adaptive Multiclass Mahalanobis Taguchi System

## Abstract

Bearings are vital components in industrial machines. Diagnosing the fault of rolling element bearings and ensuring normal operation is essential. However, the faults of rolling element bearings under variable conditions and the adaptive feature selection has rarely been discussed until now. Thus, it is essential to develop a practicable method to put forward the disposal of the fault under variable conditions. Considering these issues, this paper uses the method based on the Mahalanobis Taguchi System (MTS), and overcomes two shortcomings of MTS: (1) MTS is an effective tool to classify faults and has strong robustness to operating conditions, but it can only handle binary classification problems, and this paper constructs the multiclass measurement scale to deal with multi-classification problems. (2) MTS can determine important features, but uses the hard threshold to select the features, and this paper selects the proper feature sequence instead of the threshold to overcome the lesser adaptivity of the threshold configuration for signal-to-noise gain. Hence, this method proposes a novel method named adaptive Multiclass Mahalanobis Taguchi system (aMMTS), in conjunction with variational mode decomposition (VMD) and singular value decomposition (SVD), and is employed to diagnose the faults under the variable conditions. Finally, this method is verified by using the signal data collected from Case Western Reserve University Bearing Data Center. The result shows that it is accurate for bearings fault diagnosis under variable conditions.

## 1. Introduction

Rolling element bearings have wide applications in industrial machines and are one of the most critical components. If faults occur in bearings, equipment could be damaged and disasters might happen consequently. Therefore, it is essential to monitor the health conditions of bearings. The analysis of vibration signals has been a hot research pot and used to detect faults of bearings. It is crucial to recognize faults occurring in bearings and avoid fatal breakdowns as early as possible. For decades, many researchers have conducted extensive research on fault diagnosis. At present, the fault diagnosis methods are divided into model-based methods and data-driven methods. The model-based methods generally build on the physics of the process, generating the residuals between the measure process variables and estimates [1], such as Hidden Markov Modeling (HMM) which is successfully applied to bearing fault detection and diagnosis [2]. Autoregressive modelling [3] also has had excellent performance in bearing fault diagnosis. The accelerated degradation testing (ADT) [4] method is useful in fault diagnosis and lifespan prediction, and reference [5] presents a new approach using observer-based residual generation with no complicated design constraints to establish the relationship between the state estimation error and the fault signal. For the data-driven method, which is based on the historical data and does not need accurate mathematic and priori knowledge, it has wide applications in fault diagnosis. For example, Bayesian network is an excellent data-driven diagnosis method [6,7,8]. Machine learning algorithms, such as K-nearest Neighbor (KNN), Deep Convolutional Neural Networks (DCNN), and auto-encoders are effective in the fault diagnosis of various bearings [9,10,11,12]. There is also decision tree [13], Support Vector Machine (SVM) [14], and wavelet transform [15]. Artificial neural network (ANN) and Trace Ratio Linear Discriminant Analysis are other methods of diagnosing the bearings fault [16,17]. In addition, the vibration signal under the various operating conditions (especially in low rotational speed) is non-stationary and non-linear, the characteristic defect frequencies move continuously with the change of rotating speed, and is the same as the bearing that is going to break down. If the bearing progresses toward failure, the nonlinear features also start to be dominated by stochastic signal. With the vibration signal measured for diagnosing the bearing fault under variable condition, it will be difficult to diagnose the fault of bearing by using the traditional methods. Reference [18] proposed a signal selection scheme based upon two order tracking techniques from complicated non-stationary operational measured vibrations. Reference [19] reviewed features extraction methods and its application on bearing vibration signal and presents an empirical study of feature extraction methods in low rotational speed. Reference [20] proposes the estimation of instantaneous speed relative fluctuation (ISRF) in a vibration speed. Reference [21] proposes Stacked Convolutional Autoencoders (SCAE) together with DCNN in stationary and non-stationary speed operation. Graph-based rebalance semi-supervised learning (GRSSL) [22], weighted self-adaptive evolutionary extreme learning machine (WSaE-ELM) [23] and Singular Spectrum Analysis [24] are effective in diagnosing the fault under variable conditions.

However, although the aforementioned methods are effective for bearing faults diagnosis, the feature selection part is often non-adaptive or unexplainable. The conventional methods mainly extract and select features manually, which relies heavily on the experts’ knowledge and experience. Since the signals acquired in the real world might be various in many different aspects, the features selected manually might be sensitive in the variations of operation conditions and import inevitable errors for fault diagnosis. The deep learning algorithms can extract features automatically and overcome this drawback. However, deep learnings are data-hungry and require plenty of training data which are hardly acquired in practice, especially the faulty data under different conditions. Besides, the features acquired by deep learnings are unexplainable. Therefore, the aforementioned methods can hardly diagnose the faults under various operation conditions in practice. In contrast, the Mahalanobis Taguchi System (MTS) is more robust than the other methods in various operating conditions [25].

MTS offers a tool to determine important features and optimize the system. It is a different form compared with the other classification methods, because this classification model of measurement scale is constructed by using the class samples. It is useful to diagnose bearing faults under various conditions because the different pattern could be identified by using the Mahalanobis distance (MD) and Taguchi method. In this paper, MD is used to calculate the distance of the correlations between the benchmark and others, and the distance could be measured without the volatility of data. The advantage of MD is that it takes into consideration the correlations between the features and this consideration is very important in pattern analysis, which is why MTS is suitable for bearing fault diagnosis under various conditions [26]. On the other hand, the Taguchi method is used to select features without manual intervention, which could improve the robustness of the algorithm. MTS also offers an effective tool for multivariate analysis [27], considering that the bearing faults can be classified according to locations, such as inner race, outer race and rolling element [28]. However, when the conventional MTS is used for bearing fault diagnosis, misclassifications might occur due to less adaptivity of the threshold configuration for signal-to-noise gain. During the feature selection, the threshold is normally set as a constant, which might result in the overfitting problem. If the threshold value is too large, some critical features might be eliminated. On the contrary, if the threshold value is too low, some useless or harmful feature might be selected. Therefore, if the threshold value does not march the training data sufficiently, misclassifications emerge.

To overcome this drawback, this paper presents a novel method named adaptive Multiclass Mahalanobis Taguchi system (aMMTS) for bearing fault diagnosis. This method employs the MTS for multi-classification by considering different conditions as different benchmarks respectively. The results are based on the minimum MDs between the data and each benchmark data, and the label of the data is determined to be consistent with the label of the benchmark data whose MD is minimum. The method can be described briefly as follows: Firstly, after the Mahalanobis space (MS) is constructed by the two-level orthogonal array of the Taguchi method, aMMTS calculates the MDs from the data to the benchmark data and obtains features’ signal-to-noise ratios (SNRs) and gain values by using two-level orthogonal array. Secondly, the features are selected adaptively by recalculating the MDs via rearranging the order of features’ gain values by ascending and descending. Therefore, the proposed method is able to select the best classification result according to the adaptive chosen sequence of features’ SNRs instead of a hard threshold. Here, the sequence of SNR is determined by a function, which selects several maximum or minimum features to calculated MDs. Finally, a set of features with the best results is selected as the final feature vector. In this method, two different sets of training samples are employed to calculate the SNRs respectively and obtain the final feature vectors respectively. By the aforementioned improvement, the proposed aMMTS is capable to overcome the drawback of the conventional MTS and prevent the over-fitting problem. Therefore, the aMMTS is insensitive to the operation conditions and can be employed for bearing fault diagnosis.

Moreover, this method is combined with variational mode decomposition (VMD) [29] and singular value decomposition (SVD) to diagnose the faults. VMD is an entirely non-recursive algorithm, and is used to decompose the signal. It has been proven that due to the characteristics of nonlinear vibration in the bearings, VMD is more efficient than empirical mode decomposition (EMD) and Fourier transform (FT) under variable condition. SVD is used to extract the features. Therefore, VMD and SVD are employed in this paper.

The rest of this article is organized as follows: Section 2 introduces the algorithms involved in this paper. Section 3 illustrates the experiments to validate the proposed method. Section 4 is the conclusions.

## 2. Methodology 

In this paper, the main steps of fault diagnosis are signal decomposition, feature extraction and fault detection. The detailed process and method are as follows:

Step1: The VMD in conjunction with wavelet denoising is employed to eliminate the noises and decompose the raw signals;

Step2: Extracting features from the decomposed signals by SVD;

Step3: The proposed aMMTS is employed for the fault diagnosis. 

The steps of this method are shown in Figure 1.

### 2.1. VMD

After the wavelet denoising is used to estimate the noise of the raw signal, this paper employs VMD to decompose non-stationary signals. VMD can decompose a signal into different simple intrinsic mode functions, whose frequency center and bandwidth are band-limited and determined by iterative searching for the optimal solution of the variational model. The constrained formula is given as:(1)$$\underset{\mu_{k,}\omega_{k}}{\min}\{\sum_{k}\left||\partial_{t}[(\delta (t)+\frac{j}{\pi t})*\mu_{k}()t]|e^{-j\omega_{k}t}\right|{|}_{2}^{2}\}s.t.\sum \mu_{k}=f$$
where $\mu_{k}$ is the sub-signals, $\omega_{k}$ represents the center frequency of sub-modes. The optimal solution can be solved as the minimization problem, which could be addressed by introducing a quadratic penalty and Lagrangian multipliers [30]:
(2)$$L(\{\mu_{k}\}\{\omega_{k}\},\lambda )=\alpha \sum_{k}\left||\partial_{t}[(\delta (t)+\frac{j}{\pi t})*\mu_{k}(t)]|e^{-j\omega_{k}t}\right|{|}_{2}^{2}+||f(t)-\sum_{k}\mu_{k}(t)|{|}_{2}^{2}+\langle \lambda (t),f(t)-\sum_{k}\mu_{k}(t)\rangle$$

α denotes the balancing parameter of the constraint.

All sub-signals $\mu_{k}$ are updated for all ω≥0 as follow:(3)$${\hat{\mu }}_{k}^{n+1}\leftarrow \frac{\hat{f}-\sum_{i<k}{\hat{\mu }}_{i}^{n+1}-\sum_{i>k}{\hat{\mu }}_{i}^{n}+\frac{{\hat{\lambda }}^{n}}{2}}{1+2\alpha {\left(\omega -\omega_{k}^{n}\right)}^{2}}$$

${\hat{\mu }}_{k}^{1}$, ${\hat{\omega }}_{k}^{1}$ and ${\hat{\lambda }}^{1}$ are initialized to all zeroes.

All center frequency of sub-modes $\omega_{k}$ are updated as follow:(4)$$\omega_{k}\leftarrow \frac{\int_{0}^{\infty }\omega {|{\hat{\mu }}_{k}^{n+1}(\omega )|}^{2}d\omega }{\int_{0}^{\infty }{|{\hat{\mu }}_{k}^{n+1}(\omega )|}^{2}d\omega }$$

End for
(5)$${\hat{\lambda }}^{n+1}\leftarrow {\hat{\lambda }}^{n}+\tau (\hat{f}-\sum_{k}{\hat{\mu }}_{k}^{n+1})$$

Until:(6)∑k‖μ^kn+1−μ^kn‖22‖μ^kn‖22<ε

### 2.2. SVD

After the signals are decomposed into several modes by VMD, the features are extracted from the modes by SVD, and can be constructed as a feature matrix. SVD is a powerful tool for feature extraction in linear algebra. According to SVD, the matrix could be decomposed as follow:(7)$$X=U\omega V^{T}$$
where *X* represents a m×n matrix. There are two orthogonal matrices: matrix U(m×m) and V(n×n), and a singular diagonal matrix ω($\omega_{ij}\neq 0$, i=j and $\omega_{11}\geq \omega_{22}\geq \cdots \geq 0$), the diagonal element $\omega_{11},\omega_{22},\cdots ,\omega_{mm}$ is the singular value of *X*. U is called left singular vector, and the columns of U. V is called right singular vector. The columns of U and V especially are orthogonal to each other, and are base vector. To obtain more intrinsic information in the matrix, the singular vectors are selected. As a consequence, SVD is employed to decompose the eigenmatrix, and obtained the singular value vectors $\left(\omega_{11},\omega_{22},\cdots ,\omega_{mm}\right)$.

### 2.3. Mahalanobis–Taguchi System

Mahalanobis–Taguchi System is a pattern recognition method integrated by the MD, orthogonal table and other tools such as SNR that are proposed by Taguchi [31], who introduces the experimental design of the field SNR to pattern recognition, which can reduce the dimensions of data, and use the orthogonal table to construct the MS. The MSs are used to calculate the MDs of the experimental data, and the valid features are distinguished by the SNR. Then, the MSs are recalculated by using the valid features. Finally, the results are obtained. The calculation of the MD is described as follows:

#### 2.3.1. Mahalanobis Distance

The MD is a method of using normal data to normalize the fault data to compute the average distance between points and groups using normal data, the calculation formula of MD is as follows:(8)$$MD_{j}=\frac{1}{k}Z_{ij}^{T}C^{-1}Z_{ij}$$
(9)$$Z_{ij}=\{z_{1j},z_{2j},z_{3j},\ldots ,z_{kj}\},z_{ij}=\frac{x_{ij}-{\bar{x}}_{i}}{s_{i}}$$

$MD_{j}$ represents the MD of the jth sample, k represents the number of the feature, $x_{ij}$ represents the ith feature’s value of the of the jth sample ${\bar{x}}_{i}$ represents the mean of ith feature, $s_{i}$ represents the Standard deviation of ith feature, $C^{-1}$ represents the inverse matrix of the correlation coefficient matrix.

#### 2.3.2. Taguchi Method

In the Mahalanobis–Taguchi System, the MD measures the deviation of the test value from the normal value. The Taguchi method is able to select the features which have a larger contribution to identifying bearing faults, and then use the selected valid features to calculate the MD; the method is as below:

##### Orthogonal Array

Selecting the appropriate two-level orthogonal array, and then the k-original features obtained by VMD and SVD, are arranged into each column of the orthogonal array. In the orthogonal array, “1” indicates that the feature is selected, “2” indicates that the feature is not selected, and a MS is generated according to each row of the orthogonal array.

##### SNR and Its Gain

The main function of signal noise ratio is to select a valid feature, the calculation formula of generating SNR $\eta_{i}$ in the i line based on the orthogonal array.
(10)$$\eta_{i}=-10\;\mathrm{lg}\frac{1}{N}\sum_{j=1}^{N}{MD}_{ij}$$

j∈[1,N] represents the number of training samples.

$\eta_{i}$ represents the recognition effects of the characteristic feature, the valid feature is selected by comparing the mean of SNR of each characteristic feature at two levels. The formula is as follow:(11)$$\bar{\eta_{j}}=\frac{\sum \eta_{i}}{m}\;j=1,2$$
(12)$$\bar{\Delta \eta_{j}}=\sum \left(\frac{\bar{\eta_{j}}}{m}\right)$$

j=1,2 represents two levels, i represents the number of rows in the MS, $\bar{\eta_{1}}$ represents that in the level of ‘1’, the recognition effects to identify abnormal conditions of using this feature. $\bar{\eta_{2}}$ represents that in the level of ‘2’, the recognition effects to identify abnormal conditions of not using this feature. If $\bar{\Delta \eta_{j}}$ represents the SNR gain, $R=\{\bar{\Delta \eta_{j}}|\bar{△\eta_{i}}>0\}$ that indicates that this feature is a valid feature, if $\bar{\Delta \eta_{j}}<0$ that indicates that this feature is not a valid feature.

#### 2.3.3. Adaptive Multiclass MTS

After SNR gain is calculated, to overcome the drawback of the conventional MTS during the threshold selection, this paper presents the adaptive multiclass MTS. There are several following improvements:

(1) Solving the multiple-classification problem. Selecting samples from each kind as the benchmark data. Then, the distances between other data and each benchmark data are calculated by MTS. Therefore, the label of the benchmark data with the minimum distance is selected as the label of the training data.

(2) Selecting the features adaptively. The feature sequence is selected several times, and the best fault recognition is the one which is minus MDs between it and benchmark. It solves the error problem caused by hard threshold selection.

(3) Avoiding the overfitting problem. Since the SNR gains are calculated by training samples, different training samples are employed to calculate the MDs and identify bearing faults, and the difference validation samples are set to validate the identified result and recalculate MDs.

This method is shown in Figure 2.

This adaptive multiclass MTS can be described as follow:

First, the data are labeled, the multi-class MSs are constructed, MDs between MSs are calculated and SNR gains are obtained. This step is shown as Figure 3; *m* is the number of samples, *n* is the number of features, and *j* is the number of the label. The training data are divided into three parts, and one of them is named as Benchmark. The data $A_{j}$ and $C_{j}$ represents one kind of data respectively (such as normal, fault of inner race, outer race, rolling element), and data B includes all kinds of data.

Second, new sequences of feature parameters are generated by the sequence of features’ SNR gains in ascending and descending order, then two collections are obtained from the above sequences based on the ascending or descending order, with the ascending and descending collection as follows:(13)$$r_{k}=(\bar{△\eta_{ki}},\ldots ,\bar{△\eta_{kj}}|\bar{△\eta_{ki}}>\bar{△\eta_{kj}},i<j,i\in [1,N])$$
(14)$$q_{k}=(\bar{△\eta_{ki}},\ldots ,\bar{△\eta_{kj}}|\bar{△\eta_{ki}}<\bar{△\eta_{kj}},i>j,i\in [1,N])$$
(15)$$R=\{r_{k}|k\in [1,N]\}$$
(16)$$Q=\{q_{k}|k\in [1,N]\}$$

This step is shown in Figure 4:

Third, and the positions of feature’s SNR gain are the same as the positions of corresponding features in the sequences, the features that are used to recalculate the MDs are selected by the corresponding feature’s SNR gain. This step is shown in Figure 5. The MDs that are between each kind of A and Ci are calculated. 

Forth, the proper sequence of SNR gain is chosen by a function, which is according to the minimum MD. If two labels of data corresponding to minimum MD are the same, the recognition result is right, and accumulate the number of right recognition results. S is the recognition result. This step is shown in Figure 6.

Finally, the recognition result is verified if there is a unique optimal recognition result, and the SNR gain’s position in the sequence is the feature’s order. If there are several optimal recognition results, repeat step 3 to recalculate the result. Afterwards, a set of sequences with the best recognition effect is determined as the feature sequence, which solves the self-adaptation problem of thresholds.

## 3. Results

In this paper, the experimental data are from Case Western Reserve University Bearing Data Center. This experiment involved three different faults that occurred on three components: inner race, outer race and rolling element. The vibration signals were acquired under four different speeds: 1797 r/min, 1772 r/min, 1750 r/min, and 1730 r/min, and the sampling frequency was set to 12 kHz. To demonstrate the aMMTS, this study randomly selected the data in the dataset under the defect of 0.07 inches. The number of samples are shown in Table 1.

There were 2192 samples; 548 for inner race, 548 for outer race, 548 for rolling element and 548 for normal. The data are divided into three parts: training data, validation data and test data. In order to avoid the overfitting caused by the training data, training samples were used to construct MS, generate the SNR gain and calculate MDs by using the sequences of SNR gains, and were divided into three parts, with one of the parts set as benchmark group. To avoid the over-fitting problem, group A was used to construct MS and generate the SNR gain, and group B were used to calculate the MDs with the sequences of SNR gain and identify faults.

Validation samples were used to verify the recognition result if there exists the same minimum MDs, and the sequence was selected according to the best result.

Test samples were used to validate the proposed method.

### 3.1. Signal Decomposition by Using VMD and Wavelet Denosing

Above all, this study employed wavelet denoising to remove the noise from the raw signals. First, the Daubechies 5 (db5) was used to decompose the signal, and obtained the wavelet decomposition vector and the bookkeeping vector. Second, thresholds wavelet coefficient was calculated by setting the detail vector which would be compressed as [1,2,3] and the vector which is the corresponding percentages of lower coefficients as [100,90,80], and using the wavelet decomposition vector and the bookkeeping vector. Lastly, the thresholds, Daubechies 5 (db5) and decomposed signals were used to reconstruct the denoising signal. Then the VMD was used to decompose the signal, and was needed to give the preset IMF component number *K* and penalty parameter *α* which constrained the moderate bandwidth. The value of *α* toke the default value 1024, the value of *K* was 8. An example is shown in Figure 7.

IMFs are as shown in Figure 8.

### 3.2. Feature Extraction by Using SVD

SVD was used to analyze the IMFs. After the signal decomposition, the IMF matrix was decomposed by SVD, and obtained singular value vectors. The singular value vectors were considered as features and formed the feature matrix. Then, the feature matrix was used to diagnose the fault by aMMTS. To avoid the over-fitting problem, the features were divided into training samples, validation samples and test samples. The features of the above IMFs of those were shown in Table 2.

The features obtained by SVD are shown in Table 3.

### 3.3. Fault Diagnosis Using aMMTS

After the feature extraction, the aMMTS was used to identify and diagnose fault modes. The steps of aMMTS are as follow:

Firstly, the MS of training and benchmark were constructed, the eight-factor and two-level orthogonal array is shown in Table 4, and the MS based on Table 2 is shown in Table 5;

Secondly, the MD was calculated, and SNR gain was also obtained by benchmark samples and training samples. The SNR gain is shown in Table 6;

Thirdly, the MDs between the benchmark samples and validation samples were calculated by using the ascending and descending order of SNR;

Fourthly, the validation samples were used to verify the correctness of feature selection which existed more than one smallest MD;

Fifthly, the best sequence was chosen and set as the sequence of features.

Lastly, the best sequence was used to identify the test samples. Took the benchmark is outer race as the example shown in Figure 9.

Finally, the test sample was used to test the result of the method, and the benchmarks were inner race, rolling element, outer race and normal. The results are shown in Table 7 and the MDs between benchmark and test sample are shown in Figure 10.

As shown in Table 7 and Figure 10, this method accurately classified and diagnosed the fault of the bearing by using the different benchmarks. The recognition results of normal and outer race reached 100%. However, it is not accurate enough to diagnose the fault of inner race and rolling element. However, in the normal and the fault of inner race, it is effective in industrial application.

## 4. Discussion

Rolling element bearings are one of the most frequently used components in rotating machineries. This paper presents the method based on the wavelet denoising VMD-SVD-aMMTS to diagnose the fault of bearings under the variable conditions. Firstly, VMD is used to decompose the signal. Secondly, SVD is used to extract the feature. The adaptive aMMTS uses the feature sequences and multi-benchmarks to overcome the drawback of MTS for adaptive feature selection, multi-classification and over-fitting. The experimental result shows that the method could accurately diagnose faults effectively.

However, in the actual situation, there is an imbalance between fault data and normal data. In this method, aMMTS lacks research on the imbalanced study. The absence of faulty data may create a new problem, such over-fitting. Therefore, additional experiments under imbalanced data should be done to improve the method.

## Figures and Tables

**Figure 1 sensors-19-00026-f001:**
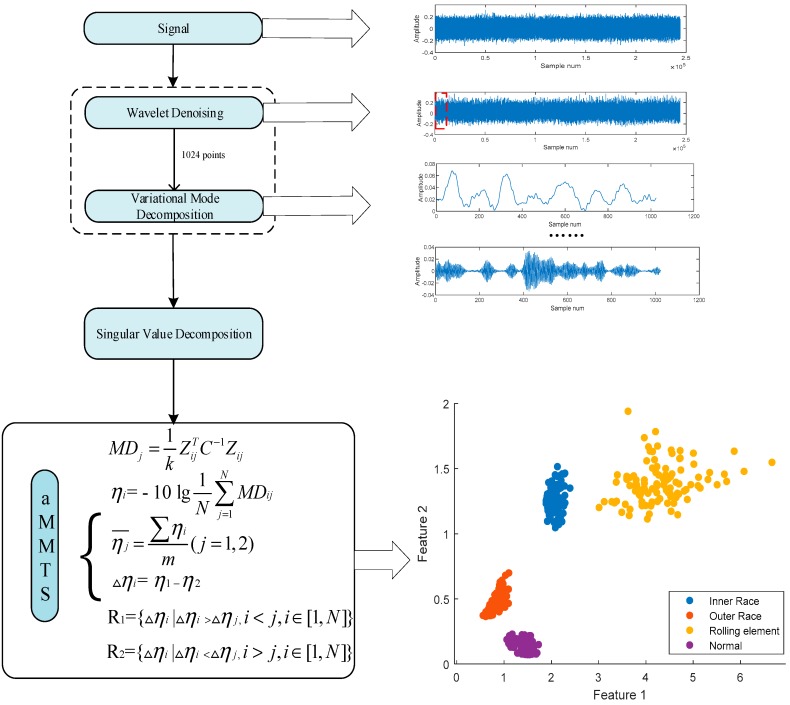
The scheme of the proposed fault diagnosis method.

**Figure 2 sensors-19-00026-f002:**
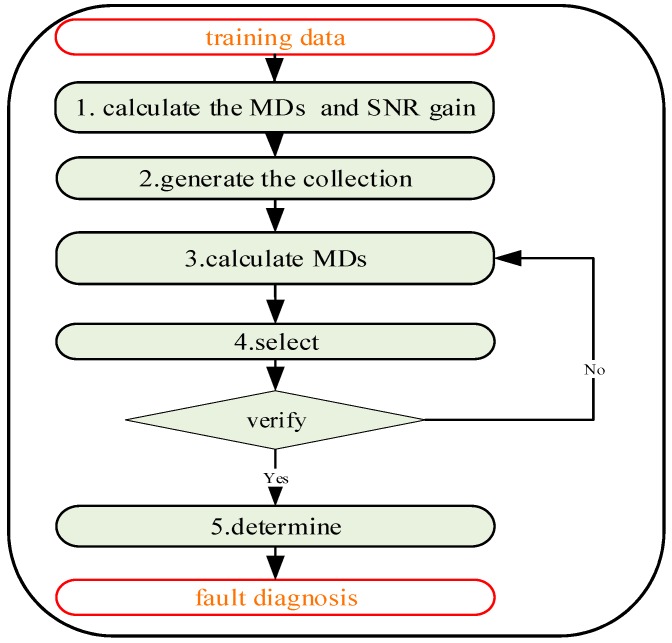
The step of Multiclass–Mahalanobis–Taguchi system (aMMTS).

**Figure 3 sensors-19-00026-f003:**
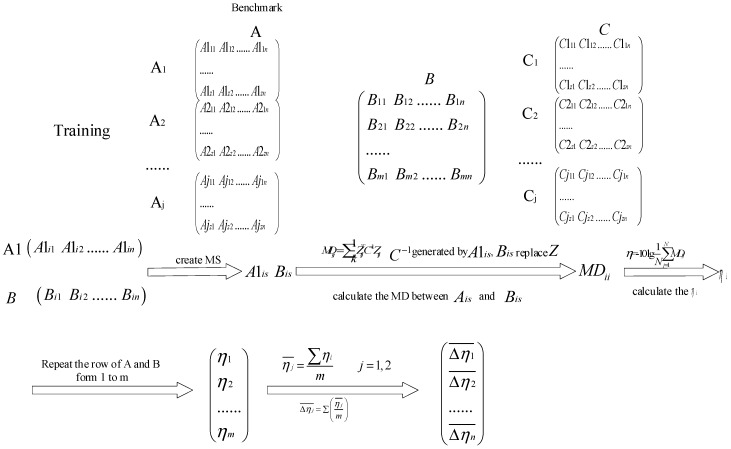
The first step of aMMTS.

**Figure 4 sensors-19-00026-f004:**
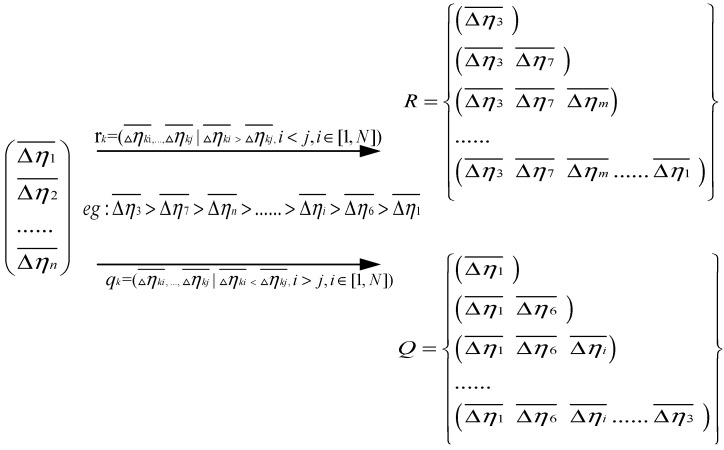
The second step of aMMTS.

**Figure 5 sensors-19-00026-f005:**
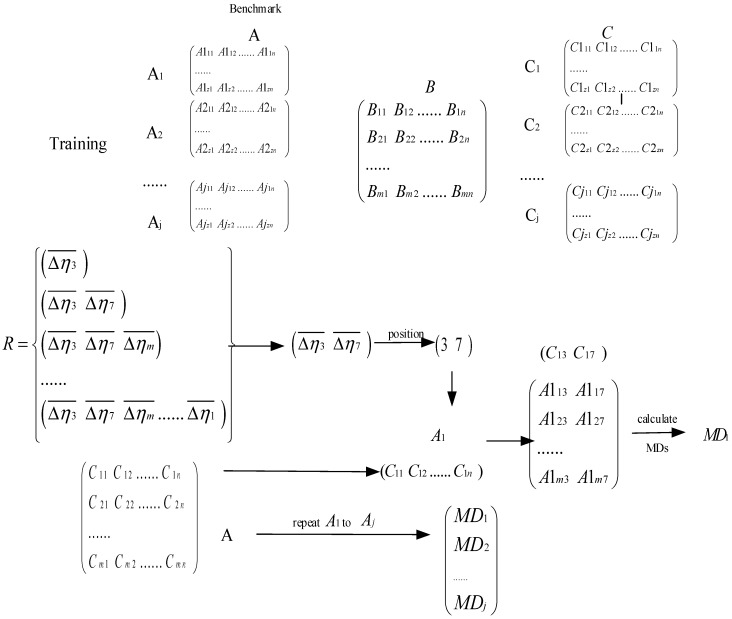
The third step of aMMTS.

**Figure 6 sensors-19-00026-f006:**
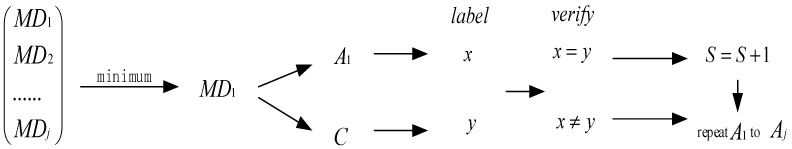
The forth step of aMMTS.

**Figure 7 sensors-19-00026-f007:**
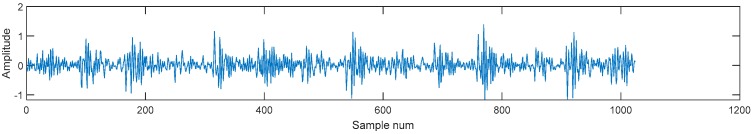
The intercepted signal.

**Figure 8 sensors-19-00026-f008:**
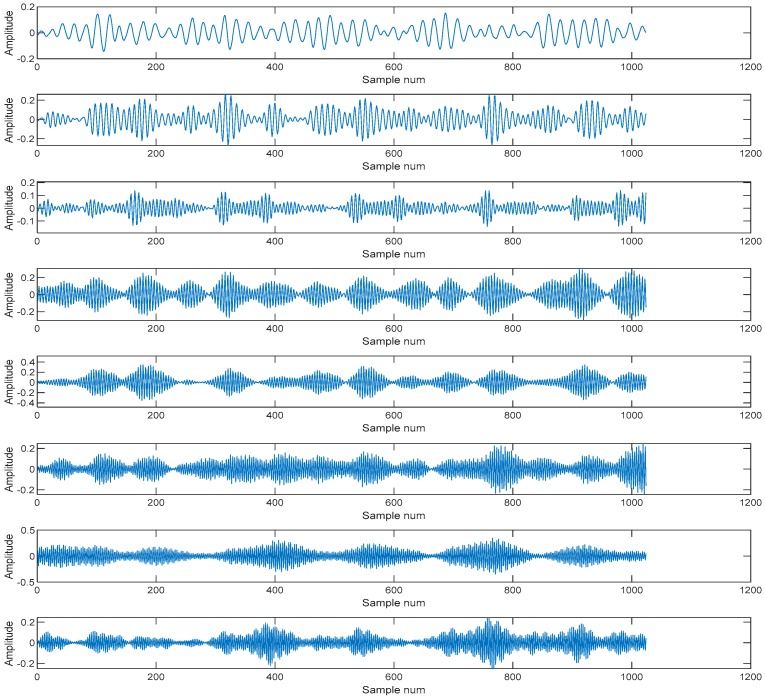
The IMFs.

**Figure 9 sensors-19-00026-f009:**
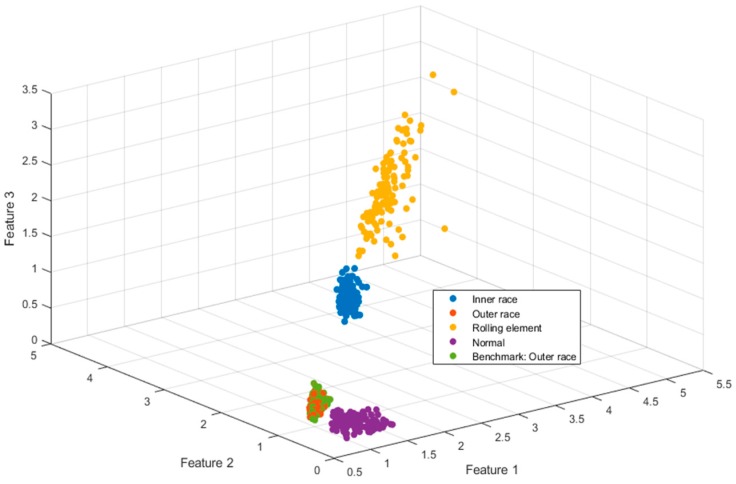
The classification result of outer race.

**Figure 10 sensors-19-00026-f010:**
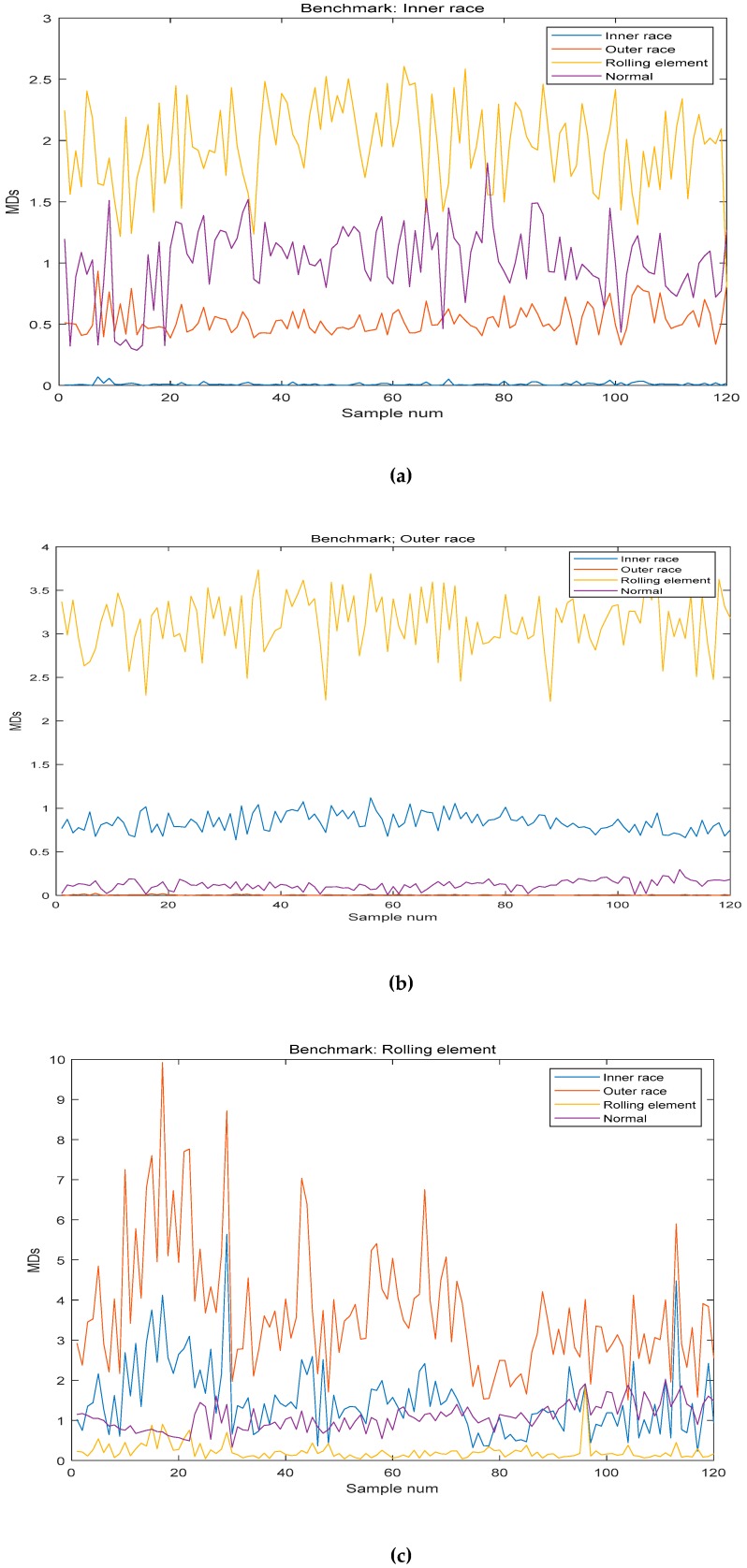
The Mahalanobis distances (MDs) between benchmark and testing data: (**a**) The MDs between benchmark (Inner race) and testing data; (**b**) The MDs between benchmark (Outer race) and testing data; (**c**) The MDs between benchmark (Rolling element) and testing data; (**d**) The MDs between benchmark (Normal) and testing data.

**Table 1 sensors-19-00026-t001:** The number of samples.

	Label	Motor Load (hp)	Speed (r/min)	Training Samples			Validation Samples	Test Samples
				BenchMark	Group A	Group B		
Inner Race	1	0 (0W)	1797	27	27	29	27	30
	1	1 (735W)	1772	27	27	29	27	30
	1	2 (1470W)	1750	27	27	29	27	30
	1	3 (2205W)	1730	27	27	29	27	30
Outer Race	2	0 (0W)	1797	27	27	29	27	30
	2	1 (735W)	1772	27	27	29	27	30
	2	2 (1470W)	1750	27	27	29	27	30
	2	3 (2205W)	1730	27	27	29	27	30
Rolling Element	3	0 (0W)	1797	27	27	29	27	30
	3	1 (735W)	1772	27	27	29	27	30
	3	2 (1470W)	1750	27	27	29	27	30
	3	3 (2205W)	1730	27	27	29	27	30
Normal	0	0 (0W)	1797	27	27	29	27	30
	0	1 (735W)	1772	27	27	29	27	30
	0	2 (1470W)	1750	27	27	29	27	30
	0	3 (2205W)	1730	27	27	29	27	30

**Table 2 sensors-19-00026-t002:** The feature of IMFs.

	IMF1	IMF2	IMF3	IMF4	IMF5	IMF6	IMF7	IMF8
Features	2.019	1.968	1.836	1.582	1.491	1.459	1.264	1.168

**Table 3 sensors-19-00026-t003:** The features of decomposed signals.

	Features							
	1	2	3	4	5	6	7	8
Normal	1.554	1.116	0.848	0.479	0.412	0.324	0.214	0.095
	1.472	1.230	0.473	0.425	0.292	0.258	0.197	0.083
	1.094	1.034	0.897	0.428	0.321	0.270	0.185	0.082
	1.173	0.939	0.931	0.683	0.387	0.287	0.229	0.080
Inner Race	2.020	1.968	1.836	1.582	1.491	1.459	1.264	1.168
	2.034	1.993	1.916	1.603	1.583	1.306	1.195	0.913
	2.115	2.023	1.918	1.763	1.570	1.410	1.376	1.143
	1.941	1.841	1.798	1.592	1.482	1.430	1.393	1.151
Rolling Element	0.811	0.793	0.667	0.577	0.562	0.542	0.451	0.223
	0.932	0.702	0.585	0.583	0.520	0.505	0.464	0.408
	0.740	0.657	0.556	0.515	0.501	0.487	0.435	0.356
	0.968	0.773	0.686	0.623	0.591	0.528	0.476	0.467
Outer Race	5.349	4.209	3.833	3.772	3.146	1.943	1.469	1.163
	5.312	4.468	4.003	3.505	2.988	2.751	1.416	1.145
	4.141	3.274	3.024	2.945	2.440	1.743	1.631	1.019
	3.334	2.901	2.560	2.382	2.036	1.697	1.236	0.941

**Table 4 sensors-19-00026-t004:** The eight-factor and two-level orthogonal array.

	A	B	C	D	E	F	G	H
1	1	1	1	1	1	1	1	1
2	1	1	1	1	1	2	2	2
3	1	1	2	2	2	1	1	1
4	1	2	1	2	2	1	2	2
5	1	2	2	1	2	2	1	2
6	1	2	2	2	1	2	2	1
7	2	1	2	2	1	1	2	2
8	2	1	2	1	2	2	2	1
9	2	1	1	2	2	2	1	2

**Table 5 sensors-19-00026-t005:** The Mahalanobis space (MS) based on the inner race.

	A	B	C	D	E	F	G	H
1	2.020	1.968	1.836	1.582	1.491	1.459	1.264	1.168
2	2.020	1.968	1.836	1.582	1.491			
3	2.020	1.968				1.459	1.264	1.168
4	2.020		1.836			1.459		
5	2.020			1.582			1.264	
6	2.020				1.491			1.168
7		1.968			1.491	1.459		
8		1.968		1.582				1.168
9		1.968	1.836				1.264	

**Table 6 sensors-19-00026-t006:** The signal-to-noise ratio (SNR) gain of features.

	Features							
	1	2	3	4	5	6	7	8
Normal	2.512	−0.471	0.390	0.726	0.789	−0.049	1.519	1.469
Inner Race	2.512	−0.471	0.390	0.726	0.789	−0.049	1.519	1.469
Rolling element	2.397	0.137	1.007	0.432	1.078	0.482	1.283	0.770
Outer Race	2.439	0.970	3.801	−0.104	0.676	2.216	1.501	−1.809

**Table 7 sensors-19-00026-t007:** The recognition result.

	Inner Race	Outer Race	Rolling Element	Normal	Total
Result	100%	99.16%	95%	100%	98.54%

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
