# Peer review of "Adaptive Multiclass Mahalanobis Taguchi System for Bearing Fault Diagnosis under Variable Conditions"

_sensors, 2018, doi:10.3390/s19010026_

Round 1
Reviewer 1 Report
The authors propose an Adaptive multiclass Mahalanobis-Taguchi system for bearing fault diagnosis considering variable conditions.
1) The methodology is not well described, in particular, section 2 is difficult to follow and should be improved. The figures 1, 2, 3, 4, 5, 6 don't explain easily the proposed methodology.
In the methodology section, it is not clear if you consider SVD in order to choose the first largest eigenvalues, and the corresponding columns of U and V. It is not clear the link between SVD and Mahalanobis-Taguchi System, what features extracted by SVD are used by Mahalanobis-Taguchi System?
2) The results appear to be poorly scientific.
- The authors claimed that to avoid over-fitting, features are divided into training and testing, but they don't explain if they used a cross-validation method and what kind.
- The considered algorithm present different signal processing tools that require a priori to set hyperparameters, for example, wavelet denoising requires to know a priori the wavelet function, the level of wavelet decomposition, the denoising method etc., VMD requires to know a priori the number of modes etc. In the experimental results, the authors must mention all parameters considered by each one algorithm.
3) The graphics quality of the figures 7,8,9,10 should be improved.
4) In literature, many works exist about bearing fault diagnosis considering variable conditions, for example considering the same benchmark, in "Statistical Spectral Analysis for Fault Diagnosis of Rotating Machines," in IEEE Transactions on Industrial Electronics, vol. 65, no. 5, pp. 4301-4310, May 2018.
doi: 10.1109/TIE.2017.2762623, the authors show that their proposed algorithm achieves 100% classification accuracy with 11 classes and SNR=-15dB irrespective of the operating conditions. So, the authors should compare their proposed method and their results with the literature.
Author Response
Replies to v1 Review Report
#1.
Comment:
The methodology is not well described, in particular, section 2 is difficult to follow and should be improved. The figures 1, 2, 3, 4, 5, 6 don't explain easily the proposed methodology.
In the methodology section, it is not clear if you consider SVD in order to choose the first largest eigenvalues, and the corresponding columns of U and V. It is not clear the link between SVD and Mahalanobis-Taguchi System, what features extracted by SVD are used by Mahalanobis-Taguchi System?
Answer:
Thanks for the reviewer’s kind reminder. We have added the following sentences in the methodology and Results.
In section 2, the adaptive multiclass MTS can be described as follow:
First, the data are labeled, the multi-class MSes are constructed, MDs between MSes are calculated and SNR gains are obtained. This step is shown as Figure 3, m is the number of samples, n is the number of features, j is the number of the label. The training data are divided into three parts, and one of themis named as Benchmark. The data represents one kind of data respectively (such as normal, fault of inner race, outer race, rolling element), and data B includes all kinds of data.
Figure 3. The first step of aMMTS
Second, new sequences of feature parameters are generated by the sequence of features’ SNR gains in ascending and descending order, then two collections are obtained from above sequences based on the ascending or descending order, the ascending and descending collection as equations (13-16).
This step is shown as Figure 4.
Figure 4. The second step of aMMTS
Third, and the positions of feature’s SNR gain are the same as the positions of corresponding feature in the sequences, the features what are used to recalculate the MDs are selected by the corresponding feature’s SNR gain. This step is shown as Figure 5. The MDs what are between each kind ofand are calculated.
Figure 5. The third step of aMMTS
Forth, the proper sequence of SNR gain is chosen by a function, which is according to the minimum MD. If two labels of data corresponding to minimum MD are the same, the recognition result is right, and accumulate the number of right recognition result.is the recognition result. This step is shown as Figure 6.
Figure 6. The forth step of aMMTS
Finally, the recognition result is verified, if there is only one best recognition result, the SNR gain’s position in the sequence is the feature’s order. If there is not only one best recognition result, and repeat step 3 to recalculate the result. Afterwards, a set of sequences with the best recognition effect is determined as the feature sequence, which solves the self-adaptation problem of thresholds.
In section 2.2, SVD is used to decompose the matrix which is composed by the decomposed signal, and obtain the singular value vectors. , (, and) is a singular diagonal matrix, and is the singular value what we need. U and V is right and left singular vector, the corresponding columns of U and V are base vector, and orthogonal to each other, but we don’t use the column of U and V.
In the section 3.2, SVD is used to processing IMFs. After the signal decomposition, the IMF matrix is decomposed by SVD to obtain singular value vectors. The singular value vectors are considered as features and formed the feature matrix. Then, the feature matrix is used to diagnose the fault by aMMTS.
#2.
Comment:
The results appear to be poorly scientific.
- The authors claimed that to avoid over-fitting, features are divided into training and testing, but they don't explain if they used a cross-validation method and what kind.
- The considered algorithm present different signal processing tools that require a priori to set hyperparameters, for example, wavelet denoising requires to know a priori the wavelet function, the level of wavelet decomposition, the denoising method etc., VMD requires to know a priori the number of modes etc. In the experimental results, the authors must mention all parameters considered by each one algorithm.
Answer:
Thanks for the reviewer’s valuable suggestion. We have rewritten sentences in line 277-279, 289-292.
The step of aMMTS is same as the step of MTS, the first step is selecting the feature based on Taguchi method, and the second step is classified the fault by Mahalanbis distance. All the training data could divide into two classes, one class is benchmark, and another is data which would be selected as features and diagnosed the fault. But in the aMMTS, the MDs are recalculating to select the best recognition, and the recognition would be verify, if only one training data are used to calculate and recalculate the MDs, the result might be overfitting. So we divide the training data into three parts: benchmark, group A and group B. There were 2192 sample, 548 for inner race, 548 for outer race, 548 for rolling element and 548 for normal. The samples are divided into three parts: training data, validation data and test data. In order to avoid the overfitting caused by the training data, therefore, the training samples are divided into three parts, one of the parts is set as benchmark group. To avoid the over-fitting problem, group A is used to construct MS and generate the SNR gain, group B are used to calculate the MDs with the sequences of SNR gain and identify faults.
This study employs the wavelet denoising to remove the noise from the raw signals. First, we use the Daubechies 5 (db5) to decompose the signal, and obtained the wavelet decomposition vector and the bookkeeping vector. Second, thresholds wavelet coefficient is calculated by setting the detail vector which would be compressed as [1, 2, 3] and the vector which is the corresponding percentages of lower coefficients as [100, 90, 80], and using the wavelet decomposition vector and the bookkeeping vector. Lastly, we use the thresholds, Daubechies 5 (db5) and decomposed signals to reconstruct the signal, and the signal was the denoising signal. Then the VMD is used to decompose signal, and needed to give the preset IMF component number K of and penalty parameter α. The value of α takes the default value 1024, the value of K is 8.
#3.
Comment:
The graphics quality of the figures 7,8,9,10 should be improved.
Answer:
Thanks for the reviewer’s kind reminder. We have updated the figures.
Figure 7. The intercepted signal
Figure 8. The IMFs
Figure 9.The classification result of outer race
Figure 10. The MDs between Benchmark and testing data
#4.
Comment:
In literature, many works exist about bearing fault diagnosis considering variable conditions, for example considering the same benchmark, in "Statistical Spectral Analysis for Fault Diagnosis of Rotating Machines," in IEEE Transactions on Industrial Electronics, vol. 65, no. 5, pp. 4301-4310, May 2018. doi: 10.1109/TIE.2017.2762623, the authors show that their proposed algorithm achieves 100% classification accuracy with 11 classes and SNR=-15dB irrespective of the operating conditions. So, the authors should compare their proposed method and their results with the literature.
Answer:
Thanks for the reviewer’s valuable suggestion. The method proposed by the mentioned acticle could reduce the computational cost of the algorithm, diagnose the fault without the training of classifier and have high accuracy and robust fault classifier with poor SNR. It’s an effective method. We have cited it in this paper. This paper has advantage in the feature selection, aMMTS could select the feature adaptively, and what’s more, Mahalanobis–Taguchi System is a method for the real-time decision making, it’s a white box in the feature selection. And Mahalanbis distance could calculate the distance between correlation variables. All of the advantages make the MTS has robust fault classifier with poor SNR and could diagnose the fault under the operating conditions, and its other advantages as follow.
• It is a robust methodology that is insensitive to variations in multidimensional systems.
• It can handle many different types of data sets and effectively consolidates the data into a useful metric.
• Implementation of MTS requires limited knowledge of statistics.
• It relies typically on simple arithmetic, contextual knowledge, and intuition.
• Its efficiency has been demonstrated in various practical applications.

Reviewer 2 Report
The paper proposed the Mahalanobis-Taguchi method for bearing fault diagnosis under variable conditions. The following are some suggestions to improve the quality of the paper:
1. The variable condition of bearing is close related to the non-stationary or transient bearing signal. This is usually occurs in low rotational speed bearings instead of high rotational speed. Since only two papers about the variable conditions, I suggests Authors provide more cited paper in Introduction about the non-stationary or transient bearing signal, for example:
- A review of feature extraction methods in vibration-based condition monitoring and its application for degradation trend estimation of low-speed slew bearing, Machines, 2017.
- A new method for the estimation of the instantaneous speed relative fluctuation in a vibration signal based on the short time scale transform, Mechanical Systems and Signal Processing, 2009.
2. Figure 1 is not obviously presented. Please revise Figure 1 with higher resolution and proper font size.
3. According to the result presented in Table 1, the bearing speed varied from 1730 to 1797. The different only about 60 rpm, is this being said by the variable conditions? If possible, Authors should also test the propose method for the bearing speed below 1000 rpm.
4. Figure 7, 8, 9 and 10 has poor resolution. Are these figure snapshot from other documents? Please presented the Figure from the MATLAB results directly.
5. The Benchmark (blue) and Outer race (orange) presented in Figure 9 is overlapping each other. If there are two classes overlapping each other, then the classification accuracy will be reduce. How the Authors can achieved into 98.54% (Table 7).
Author Response
Replies to v2 Review Report
#1.
Comment:
The variable condition of bearing is close related to the non-stationary or transient bearing signal. This is usually occurs in low rotational speed bearings instead of high rotational speed. Since only two papers about the variable conditions, I suggests Authors provide more cited paper in Introduction about the non-stationary or transient bearing signal, for example:
- A review of feature extraction methods in vibration-based condition monitoring and its application for degradation trend estimation of low-speed slew bearing, Machines, 2017.
- A new method for the estimation of the instantaneous speed relative fluctuation in a vibration signal based on the short time scale transform, Mechanical Systems and Signal Processing, 2009.
Answer:
Thanks for the reviewer’s valuable suggestion. We have cited the related references supporting the statement in.
#2.
Comment:
Figure 1 is not obviously presented. Please revise Figure 1 with higher resolution and proper font size.
Answer:
Thanks for the reviewer’s kind reminder. We have updated the figure.
Figure 1. The scheme of the proposed fault diagnosis method
#3.
Comment:
According to the result presented in Table 1, the bearing speed varied from 1730 to 1797. The different only about 60 rpm, is this being said by the variable conditions? If possible, Authors should also test the propose method for the bearing speed below 1000 rpm.
Answer:
Thanks for the reviewer’s valuable suggestion.
The data from the Bearing Data Center of Case Western Reserve University is provided the bearing speed varied from 1730 to 1797 r/min. And we add the list of load range from 0 to 3 horsepower (0 to 2205W) in the Table 1. The invovled data is consisted of four operating condition with the load range from 0 to 3 horsepower and speed range from 1797 to 1720 RPM respectively. If we get the bearing speed below 1000 rpm, we will use it to test and perfect the propose method.
Table 1. The number of samples
label | Motor load(hp) | Speed (r/min) | Training samples | Validation samples | Test samples | |||
Benchmark | Group A | Group B | ||||||
Inner race | 1 | 0 (0W) | 1797 | 27 | 27 | 29 | 27 | 30 |
1 | 1 (735W) | 1772 | 27 | 27 | 29 | 27 | 30 | |
1 | 2 (1470W) | 1750 | 27 | 27 | 29 | 27 | 30 | |
1 | 3 (2205W) | 1730 | 27 | 27 | 29 | 27 | 30 | |
Outer race | 2 | 0 (0W) | 1797 | 27 | 27 | 29 | 27 | 30 |
2 | 1 (735W) | 1772 | 27 | 27 | 29 | 27 | 30 | |
2 | 2 (1470W) | 1750 | 27 | 27 | 29 | 27 | 30 | |
2 | 3 (2205W) | 1730 | 27 | 27 | 29 | 27 | 30 | |
Rolling element | 3 | 0 (0W) | 1797 | 27 | 27 | 29 | 27 | 30 |
3 | 1 (735W) | 1772 | 27 | 27 | 29 | 27 | 30 | |
3 | 2 (1470W) | 1750 | 27 | 27 | 29 | 27 | 30 | |
3 | 3 (2205W) | 1730 | 27 | 27 | 29 | 27 | 30 | |
Normal | 0 | 0 (0W) | 1797 | 27 | 27 | 29 | 27 | 30 |
0 | 1 (735W) | 1772 | 27 | 27 | 29 | 27 | 30 | |
0 | 2 (1470W) | 1750 | 27 | 27 | 29 | 27 | 30 | |
0 | 3 (2205W) | 1730 | 27 | 27 | 29 | 27 | 30 | |
#4.
Comment:
Figure 7, 8, 9 and 10 has poor resolution. Are these figure snapshot from other documents? Please presented the Figure from the MATLAB results directly.
Answer:
Thanks for the reviewer’s kind reminder. We have updated the Figure from the MATLAB results directly.
Figure 7. The intercepted signal
Figure 8. The IMFs
Figure 9.The classification result of outer race
Figure 10. The MDs between Benchmark and testing data
#5.
Comment:
The Benchmark (blue) and Outer race (orange) presented in Figure 9 is overlapping each other. If there are two classes overlapping each other, then the classification accuracy will be reduce. How the Authors can achieved into 98.54% (Table 7).
Answer:
Thanks for the reviewer’s valuable suggestion. The fault type of Benchmark (blue) which is illustrated in the Figure 9 is Outer race, thus it should be overlapping with the Outer race (orange), and we present the classification of the Inner race.
Figure. The type of Benchmark is Inner race

Reviewer 3 Report
This paper proposes a novel method named adaptive Multiclass–Mahalanobis–Taguchi system (aMMTS), in conjunction with variational mode decomposition (VMD) and singular value decomposition (SVD) to diagnosis the faults under the variable conditions. The paper is okay, some important issues should be explained:
1. It is good to classify the fault diagnosis methods, such as data-driven, model-based, in the introduction, not to put them together. Please update the introduction, and made it clearly.
2. Have you compare the proposed method with the existing methods for bearing? I think the fault diagnosis accuracy is most important, please state the issue.
3. What are the effects of speed of bearing on the diagnosis performance?
4. Some similar FD method could be referred to, such as “A data-driven fault diagnosis methodology in three-phase inverters for PMSM drive systems”, “Bayesian Networks in Fault Diagnosis” and “Application of Bayesian networks in reliability evaluation”.
5. Fig 1 is not clear. Pls update it, especially the sub-fig.
Author Response
Replies to v3 Review Report
#1.
Comment:
It is good to classify the fault diagnosis methods, such as data-driven, model-based, in the introduction, not to put them together. Please update the introduction, and made it clearly.
Answer:
Thanks for the reviewer’s kind reminder. We have updated the introduction.
#2.
Comment:
Have you compare the proposed method with the existing methods for bearing? I think the fault diagnosis accuracy is most important, please state the issue.
Answer:
Thanks for the reviewer’s valuable suggestion. The fault diagnosis accuracy is important, but some methods like CNN have high accuracy, but it’s a black box module, we don’t how and why it produce certain outputs, it is highly unpredictable, and it’s not real-time method, so this method is not suitable for the application in the fault diagnose of real-time or precision instruments.
Mahalanobis–Taguchi System is a method for real-time decision making and has advantage in feature selection. It’s a white box module, especially in the feature selection, and MTS has advantages as follow.
• It is a robust methodology that is insensitive to variations in multidimensional systems.
• It can handle many different types of data sets and effectively consolidates the data into a useful metric.
• Implementation of MTS requires limited knowledge of statistics.
• It relies typically on simple arithmetic, contextual knowledge, and intuition.
• Its efficiency has been demonstrated in various practical applications.
#3.
Comment:
What are the effects of speed of bearing on the diagnosis performance?
Answer:
Thanks for the reviewer’s kind reminder. When the rotating speed and the load of a bearing are constant, the fault of the bearing can be easily diagnosed. But the variation of bearing’s speed will cause non-stationary and non-linear vibration, the vibration signal will also be non-stationary and non-linear, the characteristic defect frequencies move continuously with the change of rotating speed. And if the bearing progresses toward failure, the nonlinear features also start to be dominated by stochastic signal. So the vibration signal measured for diagnosing the bearing fault under variable condition, it will be difficult to diagnose the fault of bearing by using the traditional methods.
#4.
Comment:
Some similar FD method could be referred to, such as “A data-driven fault diagnosis methodology in three-phase inverters for PMSM drive systems”, “Bayesian Networks in Fault Diagnosis” and “Application of Bayesian networks in reliability evaluation”.
Answer:
Thanks for the reviewer’s valuable suggestion. We have added related references supporting the statement.
#5.
Comment:
Fig 1 is not clear. Pls update it, especially the sub-fig
Answer:
Thanks for the reviewer’s kind reminder. We have updated the figure.
Figure 1. The scheme of the proposed fault diagnosis method

Round 2
Reviewer 1 Report
The authors improved the manuscript describing in detail the technical aspects of the proposed algorithm. The introduction and the literature review have been improved as well.
Reviewer 2 Report
Dear Authors, Thank you for the effort in revising the paper. The paper is now improved significantly.
Reviewer 3 Report
It is okay with the revision.